# Sigma-Pi Structure with Bernoulli Random Variables: Power-Law Bounds for Probability Distributions and Growth Models with Interdependent Entities

**DOI:** 10.3390/e23020241

**Published:** 2021-02-19

**Authors:** Arthur Matsuo Yamashita Rios de Sousa, Hideki Takayasu, Didier Sornette, Misako Takayasu

**Affiliations:** 1Institute of Innovative Research, Tokyo Institute of Technology, Yokohama 226-8502, Japan; yamashita.a.ai@m.titech.ac.jp; 2Sony Computer Science Laboratories, Tokyo 141-0022, Japan; takayasu@csl.sony.co.jp; 3Department of Management, Technology and Economics, ETH Zürich, 8092 Zürich, Switzerland; dsornette@ethz.ch; 4Swiss Finance Institute, University of Geneva, 1211 Geneva, Switzerland; 5Department of Mathematical and Computing Science, School of Computing, Tokyo Institute of Technology, Yokohama 226-8502, Japan

**Keywords:** probability theory, power-law, stochastic processes, random multiplicative process, growth model

## Abstract

The Sigma-Pi structure investigated in this work consists of the sum of products of an increasing number of identically distributed random variables. It appears in stochastic processes with random coefficients and also in models of growth of entities such as business firms and cities. We study the Sigma-Pi structure with Bernoulli random variables and find that its probability distribution is always bounded from below by a power-law function regardless of whether the random variables are mutually independent or duplicated. In particular, we investigate the case in which the asymptotic probability distribution has always upper and lower power-law bounds with the same tail-index, which depends on the parameters of the distribution of the random variables. We illustrate the Sigma-Pi structure in the context of a simple growth model with successively born entities growing according to a stochastic proportional growth law, taking both Bernoulli, confirming the theoretical results, and half-normal random variables, for which the numerical results can be rationalized using insights from the Bernoulli case. We analyze the interdependence among entities represented by the product terms within the Sigma-Pi structure, the possible presence of memory in growth factors, and the contribution of each product term to the whole Sigma-Pi structure. We highlight the influence of the degree of interdependence among entities in the number of terms that effectively contribute to the total sum of sizes, reaching the limiting case of a single term dominating extreme values of the Sigma-Pi structure when all entities grow independently.

## 1. Introduction

In [1], we introduced the Sigma-Pi structure with identically distributed random variables as
(1)$$X_{n}=c_{0}+\sum_{j=1}^{n}c_{j}\prod_{k=1}^{j}\xi \left(\omega_{jk}\right),$$
where n≥1, $c_{j}$ represents real coefficients and $\xi (\omega_{jk})$ are identically distributed random variables, with possible deterministic values of labels $\omega_{jk}$ in $\left\{1,2,...,\frac{1}{2}n\left(n+1\right)\right\}$ so that variables with distinct labels are mutually independent and the ones with equal label represent repeated variables, having the same value with probability 1. Naturally, such Sigma-Pi structure—whose name was given in reference to the mathematical symbols of sum and product that appear in its definition—is itself a random variable and its convergence depends on the specification of the distribution of variables ξ and the coefficients $c_{j}$. For n→∞, we omit the index *n* and identify it just by *X*.

A first inspiration for proposing the Sigma-Pi structure is the (univariate) stochastic recurrence equation:(2)s(t)=ξ(t)s(t−1)+ζ(t),
where ξ(t) and ζ(t) are random variables, whose solution has the following Sigma-Pi structure (take for simplicity s(0)=ζ(t)=c, ∀t, and *c* real constant):(3)$$s\left(t\right)=c+\sum_{j=1}^{t}c\prod_{k=1}^{j}\xi \left(t-j+k\right);$$
observe that here $\omega_{jk}=t-j+k$ and we have $\omega_{jk}=\omega_{j^{\prime }k^{\prime }}$ for $k-j=k^{\prime }-j^{\prime }$.

Equation (Equation 2) has been extensively studied (see in [2]) and one of its most interesting features, under general conditions, is the heavy-tailed—in fact, power-law-tailed—probability distribution for s(t) even for light-tailed variables ξ(t) and ζ(t), a result proven by Kesten and Goldie [3,4]. Aside the pure mathematical interest, this equation appears in stochastic processes with random coefficients [5,6,7,8], particularly in the ARCH/GARCH processes ((Generalized) Autoregressive Conditional Heteroskedasticity) for the modeling of market price dynamics [9,10,11,12].

Being referred as random multiplicative process or Kesten process [13,14,15], Equation (Equation 2) and its variations also appear in the modeling of growth of entities, either biological populations or in social context, e.g., companies and cities sizes [16,17,18,19,20]. The presence of the Kesten process in such growth models is explained by two basic ingredients: the Gibrat’s law of proportional growth—the growth of an entity is proportional to its current size but with stochastic growth rates independent of it [21]—and a surviving mechanism to prevent the collapse to zero [14]. One of the simplest surviving mechanism is the introduction of an additive term to the basic Gibrat’s law, which results in Equation (Equation 2).

Equation (Equation 3), derived from the Kesten process, is only a specific case of the general Sigma-Pi structure, where random variables ξ of same label are repeated in different product terms (for example, variable ξ(t) occurs in all product terms). From the definition of the Sigma-Pi structure in Equation (Equation 1), numerous other configurations are possible. Mikosch et al. studied the total independence case, with each random variable ξ of a given label occurring only once, meaning all involved random variables are independent, and proved that it also has a power-law-tailed probability distribution [22]. In [1], we also considered a mixture between the total independence and the Kesten case; based on numerical simulations and heuristic arguments, we conjectured that the Sigma-Pi structure presents power-law-tailed probability distribution of same tail-index for any case provided there is no repeated variable within a single product term. Those observations suggest that the power-law tail is a general feature of the Sigma-Pi structure, with the Kesten process being a particular instance. The study of such structure can then contribute to our understanding of the emergence of power-law-tailed probability distributions, in a similar way that the central limit theorem explains the normal distribution arising from the sum of random variables.

Here, we explore the Sigma-Pi structure defined in Equation (Equation 1) with Bernoulli random variables. The investigation of the simple Bernoulli random variables in the next section allows us to obtain exact results for the general Sigma-Pi structure, a particular one being that, for any configuration of random variables, it has a distribution bounded from below by a power-law function. Focusing on the case with no repeated variable within a single product term, we further show that the asymptotic probability distribution of the Sigma-Pi structure always lies between two power-laws of same tail-index. In Section 3, we place the Sigma-Pi structure in the context of a prototypical model for growth of entities which we proposed in [1]; we illustrate the results obtained in the previous section by examining entities possibly growing interdependently and present an elementary example of temporal dependence. Despite the simplicity of the Bernoulli random variable, its study provides useful insights for other distributions, such as for half-normally distributed random variables that we analyze in Section 4 also in the growth model context and derive numerical results analogous to the Bernoulli case. We summarize and make our final comments in Section 5.

## 2. Sigma-Pi Structure with Bernoulli Random Variables

### 2.1. General Result

In this section, we study the Sigma-Pi structure with $c_{0}=0$, $c_{j}=1$, ∀j>0, and $\xi \left(\omega_{jk}\right)=\theta \eta \left(\omega_{jk}\right)$, η∼Bernoulli(p), i.e., η=1 with probability *p* and η=0 with probability 1−p. We then can write Equation (Equation 1) as
(4)$$X_{n}=\sum_{j=1}^{n}\theta^{j}Y_{j},$$
where $Y_{j}=\prod_{k=1}^{j}\eta \left(\omega_{jk}\right)$, consequently having $Y_{j}∼Bernoulli\left(p^{\gamma (j)}\right)$, not necessarily mutually independent due to the possible repetition of random variables η, with γ(j)∈{1,2, … ,j} being the number of distinct/independent random variables η in the product term $Y_{j}$.

There are two kinds of dependence due to the repetition of variables η (identified by labels $\omega_{jk}$):(a)within-dependence: repeated variables η within a single product term $Y_{j}$, i.e., $\omega_{jk}=\omega_{jk^{\prime }},k\neq k^{\prime }$. For example, in $Y_{3}=\eta \left(1\right)\eta \left(1\right)\eta \left(2\right)$ the variable η(1) appears twice ($\omega_{31}=\omega_{32}=1$). In the extreme case in which all product terms have all variables η distinct, we say that the Sigma-Pi structure has within-independence;(b)between-dependence: repeated variables η in different product terms $Y_{j}$ and $Y_{j^{\prime }}$, i.e., $\omega_{jk}=\omega_{j^{\prime }k^{\prime }},j\neq j^{\prime }$. For example, $Y_{2}=\eta \left(1\right)\eta \left(2\right)$ and $Y_{3}=\eta \left(2\right)\eta \left(3\right)\eta \left(4\right)$ share the variable η(2) ($\omega_{22}=\omega_{31}=2$). In the extreme case in which any pair of product terms have all variables η distinct, we say that the Sigma-Pi structure has between-independence.

We show here that, for any kind of within- and between-dependence, the complementary cumulative probability distribution $P(X_{n}\geq x)$ is bounded from below by a power-law function. Imposing the condition $\theta^{j}>\sum_{k=1}^{j-1}\theta^{k},\forall j\Rightarrow \theta >2$ or θ≥2, if *j* finite (see Appendix A for case θ>1), we have that the probability that $X_{n}$ is greater than or equal to $\theta^{j}$ is the complementary probability that $X_{n}$ is less than $\theta^{j}$, which happens when all terms $\theta^{k}Y_{k}$ that could be greater than or equal to $\theta^{j}$ are zero, that is,
(5)$$\begin{array}{cc} P(X_{n}\geq x=\theta^{j}) & =1-P\left(\overset{n}{\underset{k=j}{⋂}}\left(Y_{k}=0\right)\right)\\ \; & =1-P\left(Y_{j}=0\right)\prod_{k=j+1}^{n}P\left(Y_{k}=0|\overset{k-1}{\underset{l=j}{⋂}}\left(Y_{l}=0\right)\right), \end{array}$$

We indicate by (η) the irreducible set of variables η generating the product terms $Y_{j}$,$Y_{j+1}$, …, $Y_{k-1}$, i.e., the set of all variables η with distinct labels in the considered product terms. The event ${⋂}_{l=j}^{k-1}\left(Y_{l}=0\right)$ (which can be thought of as a “macrostate”, in the language of Statistical Physics) is equivalent to the union of all possible values of (η) that satisfy the condition of the event (“microstates” corresponding to the “macrostate”). We denote the set of all such (η) by {(η)}. For example, the irreducible set of variables η generating the product terms $Y_{2}$ and $Y_{3}$, with $Y_{2}=\eta \left(1\right)\eta \left(2\right)$ and $Y_{3}=\eta \left(2\right)\eta \left(2\right)\eta \left(3\right)$, is (η)=(η(1),η(2),η(3)); if the “macrostate” is $\left(Y_{2}=0\right)\cap \left(Y_{3}=0\right)$, the set of all “microstates” is {(η)}={(0,0,0),(0,0,1),(0,1,0),(1,0,0),(1,0,1)}. Then, the terms in the product in Equation (Equation 5) can be written as
(6)$$\begin{array}{cc} P\left(Y_{k}=0|\overset{k-1}{\underset{l=j}{⋂}}\left(Y_{l}=0\right)\right) & =P\left(Y_{k}=0|\underset{\left\{(\eta )\right\}}{⋃}\left(\eta \right)\right)\\ \; & =\frac{P\left({⋃}_{\left\{(\eta )\right\}}\left(\left(Y_{k}=0\right)\cap \left(\eta \right)\right)\right)}{P\left({⋃}_{\left\{(\eta )\right\}}\left(\eta \right)\right)}. \end{array}$$

As events in {(η)} are disjoint, so are events $\left(\left(Y_{k}=0\right)\cap \left(\eta \right)\right),\left(\eta \right)\in \left\{\left(\eta \right)\right\}$, and the numerator of Equation (Equation 6) reads as
(7)$$P\left(\underset{\left\{(\eta )\right\}}{⋃}\left(\left(Y_{k}=0\right)\cap \left(\eta \right)\right)\right)=\sum_{\left\{(\eta )\right\}}P\left(Y_{k}=0|\left(\eta \right)\right)P\left(\left(\eta \right)\right).$$

Observing each term in the sum in Equation (Equation 7), we have the bounds $P\left(Y_{k}=0\right)\leq P\left(Y_{k}=0|\left(\eta \right)\right)\leq 1$. They corresponds to two extreme cases: (lower) $Y_{k}$ does not share any variable η with any $Y_{l}$, j≤l≤k−1, and (upper) $Y_{k}$ shares *l* variables η with some $Y_{l}$, j≤l≤k−1. Thus, the terms in the product in Equation (Equation 5) also have the bounds
(8)$$P\left(Y_{k}=0\right)\leq P\left(Y_{k}=0|\overset{k-1}{\underset{l=j}{⋂}}\left(Y_{l}=0\right)\right)\leq 1.$$

Using Equation (Equation 8) in Equation (Equation 5) and reminding that $Y_{j}∼Bernoulli\left(p^{\gamma (j)}\right)$, we arrive at
(9)$$p^{\gamma (j)}\leq P\left(X_{n}\geq x=\theta^{j}\right)\leq 1-\prod_{k=j}^{n}\left(1-p^{\gamma (k)}\right).$$

Considering all possible within-dependence cases (reflected in the function γ), the minimum value for the lower bound in Equation (Equation 9) occurs for within-independence: γ(j)=j, and the maximum value for the upper bound occurs for total within-dependence: γ(k)=1. Then,
(10)$$p^{j}\leq P\left(X_{n}\geq x=\theta^{j}\right)\leq 1-{\left(1-p\right)}^{n-j+1}.$$

Defining $\alpha =-\frac{\log p}{\log\theta }$ such that $p^{j}=x^{-\alpha }$ for $x=\theta^{j}$ and letting n→∞ (changing notation $X_{n}$ to *X*):(11)$$x^{-\alpha }\leq P\left(X\geq x=\theta^{j}\right)\leq 1.$$

For other values of *x*, we use the property of the complementary cumulative probability:(12)$$P\left(X\geq x=\theta^{j}\right)\geq P\left(X\geq x;\theta^{j}\leq x\leq \theta^{j+1}\right)\geq P\left(X\geq x=\theta^{j+1}\right),$$
so that if $P\left(X\geq x=\theta^{j}\right)=rx^{-\alpha }$, then $r\theta^{-\alpha }x^{-\alpha }\leq P\left(X\geq x\right)\leq r\theta^{\alpha }x^{-\alpha }$ (see graphical derivation in Figure 1: values $x=\theta^{j}$ and the corresponding probabilities $P(X\geq x=\theta^{j})$ define regions where the probabilities for other values of *x* must occur).

Therefore, for any within- and between-dependence, we obtain (observing that $\theta^{-\alpha }=p$, for $\alpha =-\frac{\log p}{\log\theta }$)
(13)$$P\left(X\geq x\right)\geq px^{-\alpha },$$
that is, the Sigma-Pi structure with Bernoulli random variables has a distribution bounded from below by a power-law regardless of the type of within- and between-dependence. Observe that such bound is also valid for $X_{n}$, but the support of the distribution is bounded when *n* is finite.

### 2.2. Within-Independence

We can improve the bounds for P(X≥x) if we specify the within-dependence. We select the within-independence, γ(j)=j, ∀j, and consider the two following between-dependence cases:(i)between-independence: $\omega_{jk}\neq \omega_{j^{\prime }k^{\prime }}$, $j\neq j^{\prime }$⇒$Y_{j}∼Bernoulli\left(p^{j}\right)$, mutually independent.
(14)$$\begin{array}{cc} P_{\left[ind\right]}^{\left(ind\right)}\left(X_{n}\geq x=\theta^{j}\right) & =1-\prod_{k=j}^{n}\left(1-p^{k}\right)\\ \; & =1-{\left(p^{j};p\right)}_{n-j+1}, \end{array}$$
where the superscript (.) indicates the nature of the between-dependence, the subscript [.] indicates the nature of the within-dependence, and ${\left(a;p\right)}_{n}=\prod_{k=0}^{n-1}\left(1-ap^{k}\right)$ is the q-Pochhammer symbol [23]. For n→∞, using the expansion ${\left(a;p\right)}_{\infty }=\sum_{k=0}^{\infty }\frac{{\left(-1\right)}^{k}p^{k(k-1)/2}}{{\left(p;p\right)}_{k}}a^{k}$, we have
(15)$$P_{\left[ind\right]}^{\left(ind\right)}\left(X\geq x=\theta^{j}\right)∼\frac{1}{1-p}x^{-\alpha }.$$Then, for large *x*, using Equation (Equation 12):
(16)$$\frac{p}{1-p}x^{-\alpha }\leq P_{\left[ind\right]}^{\left(ind\right)}\left(X\geq x\right)\leq \frac{1}{p(1-p)}x^{-\alpha }.$$(ii)Kesten between-dependence: $\omega_{jk}=\omega_{(j-1)k}$, ∀j>1,k⇒$Y_{j}=Y_{j-1}\eta \left(\omega_{jj}\right)$, ∀j>1.Possible values of *X* are $X=\sum_{k=1}^{j}\theta^{k}$, with $P_{\left[ind\right]}^{\left(kes\right)}\left(X\geq x=\sum_{k=1}^{j}\theta^{k}\right)=P_{\left[ind\right]}^{\left(kes\right)}\left(X\geq x=\theta^{j}\right)=p^{j}$. Then,
(17)$$P_{\left[ind\right]}^{\left(kes\right)}\left(X\geq x=\theta^{j}\right)=x^{-\alpha },$$
and (using Equation (Equation 12))
(18)$$px^{-\alpha }\leq P_{\left[ind\right]}^{\left(kes\right)}\left(X\geq x\right)\leq \frac{1}{p}x^{-\alpha }.$$Note that for Kesten between-dependence there are restrictions on possible within-dependence: γ(1)=1; γ(j)=γ(j−1)+δ(j), δ(j)∈{0,1}, ∀j>1. For within-independence (δ(j)=1, ∀j>1), it reproduces the solution of the Kesten process.

Now observe that, for within-dependence compatible with Kesten between-dependence, in particular within-independence, the two between-dependence cases above correspond to the bounds of Equation (Equation 8): (i) $P^{\left(ind\right)}\left(Y_{k}=0|{⋂}_{l=j}^{k-1}\left(Y_{l}=0\right)\right)=P(Y_{k}=0$), because $Y_{k}$ and $Y_{l}$, j≤l≤k−1, are independent, and (ii) $P^{\left(kes\right)}\left(Y_{k}=0|{⋂}_{l=j}^{k-1}\left(Y_{l}=0\right)\right)=1$, because $Y_{k}$ contains all variables η in $Y_{l}$, j≤l≤k−1. Therefore,
(19)$$P^{\left(kes\right)}\left(X\geq x=\theta^{j}\right)\leq P\left(X\geq x=\theta^{j}\right)\leq P^{\left(ind\right)}\left(X\geq x=\theta^{j}\right).$$

From bounds in Equations (Equation 16) and (Equation 18), for within-independence and large *x*:(20)$$px^{-\alpha }\leq P_{\left[ind\right]}\left(X\geq x\right)\leq \frac{1}{p(1-p)}x^{-\alpha },$$
i.e., given within-independence, probabilities $P_{\left[ind\right]}\left(X\geq x\right)$ for large *x* always lie between power-laws with the same tail-index $\alpha =-\frac{\log p}{\log\theta }$, regardless of the between-dependence (note that α follows Kesten’s relation $\langle |\xi {|}^{\alpha }\rangle =1$ [2,3,4]). A similar result in the context of the Kesten process is known for Kesten between-dependence and random variables whose logarithms have arithmetic distributions (conditioned to non-zero values) [24], which is the case of Bernoulli random variables; here, we demonstrate that it is valid for any between-dependence case, although restricted to the Bernoulli case.

### 2.3. Max-Pi Structure

The above results for Sigma-Pi can be extended to related structures. An example is the Max-Pi structure:(21)$$X_{n}^{max}=\max\left\{c_{0},\max_{1\leq j\leq n}\left[c_{j}\prod_{k=1}^{j}\xi \left(\omega_{jk}\right)\right]\right\}.$$

For Bernoulli random variables, with the same settings we used for the Sigma-Pi structure, it reads
(22)$$X_{n}^{max}=\max_{1\leq j\leq n}\left(\theta^{j}Y_{j}\right),$$
with possible values $X_{n}^{max}=\theta^{j}$. For θ>2, we have that $P\left(X_{n}^{max}\geq x=\theta^{j}\right)=P\left(X_{n}\geq x=\theta^{j}\right)$ and previous results hold. The Max-Pi structure has potential applications in the theory of records [25].

## 3. Growth Model with Bernoulli Random Variables

### 3.1. Growth-or-Death Model

We now revisit the growth model proposed in [1] based on a simplification of growth mechanisms considered in [26,27,28], in the special case of successive births of new entities. It consists of a set of entities of sizes $s_{j}$ evolving according to Gibrat’s law and one entity born with initial size $c_{j}$ at each time step. Each entity size $s_{j}$, j≥1, at time *t* is given by
(23)sj(t)=0;at<j−1,cj;at=j−1,ξ(ωjt)sj(t−1);at≥j.

The solution for t≥j:(24)$$s_{j}\left(t\right)=c_{j}\prod_{k=j}^{t}\xi \left(\omega_{jk}\right).$$

The sum X(t) of sizes of all existing entities at time *t*, excluding the new-born entity (which only corresponds to a constant term), has a Sigma-Pi structure:(25)$$\begin{array}{cc} X(t) & =\sum_{j=1}^{t}s_{j}\left(t\right)\\ \; & =\sum_{j=1}^{t}c_{j}\prod_{k=1}^{j}\xi \left(\omega_{jk}\right) \end{array}$$
(note the index transformations: j→t−j+1 and k→t−k+1). The sum of sizes of entities is the time-evolving quantity of interest for which we seek the stationary distribution, and it has different meaning depending on the investigated system, representing, for example, “the total capitalization of a country, when entities are firms, or the total biomass of an ecosystem for biological populations” [1].

By taking Bernouli random variables $\xi \left(\omega_{jk}\right)=\theta \eta \left(\omega_{jk}\right)$, η∼Bernoulli(p), we generate the Growth-or-Death model: entities either grow by a factor θ or die. Furthermore, by choosing $c_{j}=1$, ∀j, we impose that new entities are born with unit size. Dependence among entities can be specified using the associated between-dependence of the Sigma-Pi structure. For the two extreme cases of between-dependence, we have the interpretations: (i) between-independence: each entity grows or dies independently of others, and (ii) Kesten between-dependence: factors affecting growth or death are shared by all entities so that existing entities grow or die all together.

Figure 2 and Figure 3 show the complementary cumulative distributions numerically constructed from time sampling of the growth process for between-independence and Kesten between-dependence, respectively, considering within-independence and different values of parameters θ and *p*. The distributions are in agreement with the theoretical results in previous section, within the bounds in Equations (Equation 16) and (Equation 18) and obeying the predicted tail-indices $\alpha =-\frac{\log p}{\log\theta }$. Note that the symbols in the figures refer to the possible values of *X*; in the case of Kesten between-dependence, $X=\sum_{k=1}^{j}\theta^{k}$, j≥1.

### 3.2. Mixed between-Dependence

In order to investigate examples of intermediary between-dependence cases, we take direct mixtures of between-independence and Kesten between-dependence. For this, we randomly assign each newborn entity to one of two groups: the independence group, in which members grow or die independently, or the Kesten dependence group, in which existing members grow or die all together. In practice, we use another Bernoulli random variable with parameter *q* to decide the membership of each entity, fixing q=1 for pure between-independence and q=0 for pure Kesten between-dependence and referring to the mixed cases as *q*-between-dependence.

Figure 4 depicts the complementary cumulative distributions from time sampling of the growth process taking within-independence and *q*-between-dependence. To check the transition from pure Kesten between-dependence (q=0) to between-independence (q=1), we choose the values q=0.1,0.25,0.5,0.75, and 0.9. Naturally, the exact values of probabilities differ for each value of parameter *q* (see zoom-in panel (b)) but bounds in Equation (Equation 20) are respected, including the same tail-index $\alpha =-\frac{\log p}{\log\theta }$ for all cases.

We point that, as the group assignment is stochastic for 0<q<1 and we perform time sampling, the obtained results are not from a pure Sigma-Pi structure but from an ensemble of *q*-between-dependence structures with weights privileging the ones having fraction *q* of independent entities. Nevertheless, the distribution from this model with two dependence groups respects the bounds established in the previous section because each member of this ensemble also obeys them.

### 3.3. Within-Dependence as Temporal Dependence

Making the correspondence between the considered growth model and the mathematical Sigma-Pi structure, the dependence among entities is associated with the between-dependence, as discussed above. Now, the other kind of dependence, the within-dependence, is related to the temporal dependence (memory) of the random variable η (∼growth factor). Exemplifying this connection, we study an instance of the class of κ-within-dependence, in which the values of all new variables η generated in a given time step are repeated in the next κ steps, respecting the constraints on the repetition of variables of the selected between-dependence. Case κ=0 corresponds to the within-independence examined previously. We focus here on the case κ=1, which can produce the following functions γ:(26)γa(j)=j2;ajieven,j+12;ajiodd;
(27)γb(j)=j2+1;ajieven,j+12;ajiodd

Both functions obey the restrictions for Kesten between-dependence: γ(1)=1; γ(j)=γ(j−1)+δ(j), ∀j>1. Therefore, we can use Equation (Equation 19) to determine bounds for the distribution $P_{\left[\kappa =1\right]}\left(X\geq x\right)$. We have that the maximum value of $P_{\left[\kappa =1\right]}^{\left(ind\right)}\left(X\geq x=\theta^{j}\right)$ occurs for $\gamma_{a}\left(j\right)$ and the minimum value of $P_{\left[\kappa =1\right]}^{\left(kes\right)}\left(X\geq x=\theta^{j}\right)$ occurs for $\gamma_{b}\left(j\right)$. Thus,
(28)$$P_{\left[\gamma_{b}\left(j\right)\right]}^{\left(kes\right)}\left(X\geq x=\theta^{j}\right)\leq P_{\left[\kappa =1\right]}\left(X\geq x=\theta^{j}\right)\leq P_{\left[\gamma_{a}\left(j\right)\right]}^{\left(ind\right)}\left(X\geq x=\theta^{j}\right),$$
resulting in (for large *x*)
(29)$$p^{2}x^{-\alpha /2}\leq P_{\left[\kappa =1\right]}\left(X\geq x\right)\leq \frac{1}{p}\frac{1+p}{1-p}x^{-\alpha /2}.$$

Values of $P_{\left[\kappa =1\right]}\left(X\geq x\right)$, *x* large, are also bounded by two power-laws with the same tail-index, but with tail-index $\frac{\alpha }{\kappa +1}=\frac{\alpha }{2}$, $\alpha =-\frac{\log p}{\log\theta }$. We can intuitively understand this result by comparing this 1-within-dependence case to the within-independence Sigma-Pi structure with random variables $\xi ={\left(\theta \eta^{\prime }\right)}^{2}=\theta^{2}\eta$, $\eta^{\prime },\eta ∼Bernoulli\left(p\right)$. Such Sigma-Pi resembles the solution of the 1-within-dependence growth process and also produces asymptotic power-law bounds with tail-index $\alpha^{\prime }=-\frac{\log p}{\log\theta^{2}}=-\frac{1}{2}\frac{\log p}{\log\theta }$.

Figure 5 illustrates the complementary cumulative distributions for 1-within-independence and *q*-between-dependence using the same parameters as Figure 4. For all *q*-between-dependence cases, the tail-index of the bounds follows the predicted value in Equation (Equation 29), being equal to half of the value for the within-independence case.

Temporal correlations in the Kesten process were studied in [29,30] by taking Gaussian random variables with exponentially decreasing autocorrelation function. The main finding was that the tail-index of the power-law-tailed distribution is inversely proportional to the correlation time, which is aligned with our results. Although this type of correlation does not fit in our definition for Sigma-Pi structure—each random variable is either independent or equal to another—the same intuition of re-normalizing the multiplicative random factors applies.

### 3.4. Contribution of Entities to the Total Sum

The picture suggested by the theoretical and numerical results above is that changing the within-dependence can alter the tail-index of the distribution power-law bounds, noticing that this type of dependence dictates the individual distribution of the product terms $Y_{j}$ being summed. However, for a fixed within-dependence, it seems that the tail-index remains the same for any between dependence (proved for within-dependence in Section 2).

In spite of having the same asymptotic power-law behavior, expressed in the unchanged tail-index, we know from [1] that “the nature of the construction of the events populating the tail differs” for each type of between-dependence due to the interaction between product terms/growing entities. We characterize this difference by using the Herfindhal index, commonly applied in finance as a measure of market concentration [31]. The Herfindhal index is also known, in statistical physics, as the participation ratio, and it is in particular used to quantify the heterogeneity of configurations in spin glasses and other ill-condensed matter systems [32]. In our growth model context, it is defined as the weighted average of the contribution of each entity to the total sum of sizes:(30)$$\begin{array}{cc} H(t) & =\sum_{j=1}^{t}{\left(\frac{s_{j}\left(t\right)}{X(t)}\right)}^{2}\\ \; & =\frac{\sum_{j=1}^{t}{\left(\theta^{j}Y_{j}\right)}^{2}}{{\left(\sum_{j=1}^{t}\theta^{j}Y_{j}\right)}^{2}}. \end{array}$$
The interesting quantity is the inverse of the Herfindahl index $H^{-1}$, which provides a measure of the number of entities actually important in the total sum. In particular, if $X\left(t\right)=\theta^{j}$, then $H^{-1}=1$, i.e., only one entity contributes to the total sum; not possible for the considered Bernoulli case, but if $s_{j}\left(t\right)=\frac{X(t)}{t}$, ∀j, then $H^{-1}$ has its maximum value $H^{-1}=t$, meaning a “democratic” contribution of all entities.

We plot in Figure 6 the inverse of the Herfindahl index $H^{-1}$ as a function of the total sum of sizes *X* obtained by time sampling the Growth-or-Death model, *t* large, for within-independence and q-between-dependence. We inspect the transition from pure Kesten between-dependence (q=0) to between-independence (q=1), taking the intermediary values q=0.25,0.5, and 0.75. For Kesten between-dependence, values of $H^{-1}$ correspond to the possible values $X=\sum_{k=1}^{j}\theta^{k}$ and converge to a constant different of 1 for large *X* (in this case, $H^{-1}∼3$, for θ=2). For intermediary cases, other values of *X* are now accessible, which reflects in a larger set of possible values of $H^{-1}$, but with the Kesten between-dependence case being an upper bound for them. As parameter *q* increases, meaning an increase in the fraction of independent entities, values of $H^{-1}$ close to the bound $H^{-1}=3$ decrease in the region of large *X* until the limiting between-independence case where $H^{-1}=1$ for large *X*, i.e., just one entity contributes significantly to the total sum *X*. Because large values of *X* correspond to the tail of the probability distribution, the asymptotic power-law behavior in the between-independence case is due to a single entity (of course not the same entity for all large *X*) while for the Kesten between-dependence case there are three entities—the three oldest living ones—significantly contributing to the total sum and thus to the tail behavior.

For the extreme between-dependence cases, we can make sense of those results by analyzing the possible values of H(X) and the associated discrete probability distribution. Because there is no ambiguity in the value of *X* when θ≥2, for each *X* there is a unique H=H(X) and then the probability P(H(X);X) of the H(X) associated with a specific *X* is equal to the probability P(X) of this *X*. Considering within-independence, we have
(i)between-independence: possible values of *X* are $X=\theta^{j}+\sum_{k\in A_{j}}\theta^{k}$, $A_{j}\subset \left\{1,2,\;\ldots \;,j-1\right\}$, and the corresponding probabilities are computed by setting $Y_{k}=1$ if $\theta^{k}$ is used to compose X and $Y_{k}=0$ if otherwise (see Equation (Equation 4)):
(31)$$\begin{array}{cc} P_{\left[ind\right]}^{\left(ind\right)}\left(X=\theta^{j}+\sum_{k\in A_{j}}\theta^{k}\right) & =P\left(Y_{j}=1\right)\left[\prod_{k\in A_{j}}P\left(Y_{k}=1\right)\right]\left[\prod_{k\in A_{j}^{C}}P\left(Y_{k}=0\right)\right]\\ \; & \times \left[\prod_{k=j+1}^{t}P\left(Y_{k}=0\right)\right]\\ \; & ={\left(p;p\right)}_{t}\frac{p^{j}}{1-p^{j}}\prod_{k\in A_{j}}\frac{p^{k}}{1-p^{k}}. \end{array}$$When θ=1 this corresponds to a Poisson-binomial distribution [33,34], and it is the probability distribution of the number of existing entities at time *t* in this independence case.The particular values $X=\theta^{j}$ give $H(X=\theta^{j})=1$. For large *X*, Herfindhal index H(X)=1 (or H(X)≈1) has a small probability but larger than other values:
(32)$$P_{\left[ind\right]}^{\left(ind\right)}\left(X=\theta^{j}+\sum_{k\in A_{j}}\theta^{k}\right)=P_{\left[ind\right]}^{\left(ind\right)}\left(X=\theta^{j}\right)\prod_{k\in A_{j}}\frac{p^{k}}{1-p^{k}},$$
so that, given the occurrence of a large *X*, the probability of H(X)=1 is larger than the probability of H(X)≠1.(ii)Kesten between-dependence: possible values of *X* are $X=\sum_{k=1}^{j}\theta^{k}$ and the corresponding probabilities are
(33)P[ind](kes)X=∑k=1jθk=pj(1−p);aj<t,pj;aj=t.Because of the restricted values of *X* in this Kesten case, the possible values of H(X) can be simply expressed as (see Equation (Equation 30))
(34)$$\begin{array}{cc} H\left(X=\sum_{k=1}^{j}\theta^{k}\right) & =\frac{\theta^{j}+1}{\theta^{j}-1}\frac{\theta -1}{\theta +1}\\ \; & ∼\frac{\theta -1}{\theta +1}. \end{array}$$If θ=2, $H^{-1}∼3$, as observed in Figure 6.

## 4. Growth Model with Half-Normal Random Variables

We conjecture that analogous results to the ones obtained for the Growth-or-Death model (and the corresponding Sigma-Pi structure with Bernoulli random variables) hold for a wider class of random variables. We cite as an initial justification the already mentioned works by Kesten and Goldie [3,4] and Mikosch et al. [22], which are related to the Sigma-Pi structure with within-independence and the two extreme cases of between-dependence: Kesten and independence, respectively. In those cases, the asymptotic power-law behavior of the Sigma-Pi distribution is present when the random variables ξ are such that there exists a positive number α satisfying $\langle |\xi {|}^{\alpha }\rangle =1$. Here, we provide additional evidence by numerically studying the same growth model of the previous section but taking half-normally distributed random variables, i.e., $\xi =|\xi^{\prime }|$, $\xi^{\prime }∼N\left(0,\theta^{2}\right)$.

A difference from the Bernoulli case is that for half-normal random variables there is no death and the number of entities increases endlessly. To make numerical simulations possible, we introduce a rule for death in the form of a threshold: entities with size $s_{j}\left(t\right)<10^{-10}$ are removed. We illustrate in Figure 7 the complementary cumulative distribution time sampled from the growth model for within-independence and between-independence and Kesten between-dependence. The tail-index is given by the Kesten’s relation $\langle |\xi {|}^{\alpha }\rangle =1$, which for half-normal distribution can be expressed implicitly as θ being a function of α [35]:(35)$$\theta ={\left[\frac{2^{\frac{\alpha }{2}}\Gamma \left(\frac{\alpha +1}{2}\right)}{\sqrt{\pi }}\right]}^{-\frac{1}{\alpha }}.$$
We show results for θ=1 (α=2), θ=1.253 (α=1), and θ=1.479 (α=0.5). It is interesting to observe that for α=2, tail values for the Kesten between-dependence are larger than for between-independence, i.e., the scale factor for the Kesten between-dependence is larger, deviating from the Bernoulli case result in Equation (Equation 19). For α=1 and α=0.5, tail for between-independence larger than for Kesten between-dependence is recovered.

We proceed with the comparison and consider the *q*-between-dependence, consisting of intermediary cases between independence (q=1) and Kesten dependence (q=0). Figure 8 shows the complementary cumulative distribution for intermediary parameter values q=0.1,0.25,0.5,0.75, and 0.9. Even with the commented inversion regarding the tails of the between-independence and Kesten between-dependence, all intermediary cases fall in between the extremes corresponding to between-independence and Kesten between-dependence, so that they also present power-law tail with same tail-index (but the scale factor varies from one extreme to another as *q* increases). These bounds associated with independence and Kesten dependence may provide a path to prove that any within-independence Sigma-Pi structures present power-law tails for a wider class of random variables, as the power-law tails for the extremes cases are already well established.

For within-dependence, we again consider the κ-within-dependence, κ=1, and show the results in Figure 9 taking *q*-between-dependence. For all values of *q*, the tail-index of the tails is half of the within-independence one, similar to the Bernoulli case and also rationalized by the renormalization of the random variables. Compare the inversion of scale factors for each *q* with Figure 8.

Finally, we characterize the number of contributing entities to the total sum using the Herfindhal index. In Figure 10, we observe the transition from pure Kesten between-dependence to between-independence. For half-normal random variables, there is the possibility of distinct values of $H^{-1}$ for the same *X* and now the Kesten between-dependence can have a diversity of value $H^{-1}$ for large *X*, opposed to a single constant; but as in the Bernoulli case, it still cannot take values $H^{-1}=1$ or close to it for *X* large because of the strong dependence between the entities (they share the same growth factors): if one entity is large, its close temporal neighbors are also large. As *q* increases, value $H^{-1}=1$ for large *X* starts materializing in the numerical constructions while larger values $H^{-1}$ for large *X* become less probable to appear in the simulations as also do small values of *X*. The limit is the between-independence, for which, following the Bernoulli case, the probability of $H^{-1}=1$ given large *X* to be produced in the simulations is much larger than of $H^{-1}\neq 1$, that is, here as well only one entity contributes to a large sum.

## 5. Final Remarks

The main part of this work was devoted to the study of the Sigma-Pi structure with Bernoulli random variables. As a general theoretical result, we showed that, for any kind of within- and between-dependence, the Sigma-Pi structure always presents a distribution bounded from below by a power-law function. As a particular result for within-independence, we demonstrated that, for any between-dependence, the complementary cumulative distribution of the corresponding Sigma-Pi is bounded by power-law functions with the same tail-index given by the Kesten’s relation $\langle |\xi {|}^{\alpha }\rangle =1$, which confirms our previous conjecture [1] for this particular case of Bernoulli random variables.

We then considered the Sigma-Pi structure as the solution of a simple growth model in which entities are successively born and the quantity of interest is the sum of their sizes. For Bernoulli random variables—Growth-or-Death model—we analyzed the interaction between entities for the cases of between-independence, Kesten between-dependence, and mixtures of both (*q*-between-dependence). Agreeing with theory, we found that the mixed cases are always in between the independence and Kesten cases and thus bounded by power-laws. When introducing temporal dependence of the growth factors through within-dependence, the tail-index was modified but it remained the same for all *q*-between-dependence cases, suggesting that, given a within-dependence, no between-dependence is strong enough to alter the asymptotic power-law behavior of the distribution of a Sigma-Pi structure. However, notwithstanding the same tail-index, the construction of the distribution tail differs for each between-dependence case: for Kesten between-dependence, which corresponds to the strongest interdependence among entities, the number of entities effectively contributing to tail events is maximum, and for between-independence, in the occasion of an extreme event, a single entity in the sum is overwhelmingly larger than all the others and it alone is responsible for a given tail event.

Taking Bernoulli random variables is one of the simplest cases but it provides hints for more general situations. We restate our conjecture that most of the results concerning the presence of a power-law tail and its connections with the types of dependence of the constitutive random variables hold for a wider class of random variables. We used half-normally distributed random variables to exemplify it. In particular, we observed that the tails of the distributions for *q*-between-dependence are always bounded by the independence and Kesten dependence cases, both already proven to have asymptotic power-law behavior with the same tail-index.

We conclude by indicating a direct generalization of the Sigma-Pi structure, the Sigma-Sigma-Pi structure:(36)$$X_{nm}=c_{0}+\sum_{j=1}^{n}\sum_{k=1}^{m}c_{jk}\prod_{l=1}^{j}\xi \left(\omega_{jkl}\right).$$

The Sigma-Sigma-Pi includes the Sigma-Pi but allows more than one product term of the same order, i.e., with same number of multiplicative random factors. In the same way that the univariate Kesten process has a solution following a Sigma-Pi structure, the components of the solution of the multivariate Kesten process can be represented as a Sigma-Sigma-Pi. In addition to multiplicative processes, it is also the solution of a generalized version of the studied growth model, with the possibility of more than one entity born at each time step. Results on the Sigma-Pi structure and its extensions, either as mathematical objects on their own or for modeling time series and growth of entities in diverse fields, can broaden our comprehension on the construction of particular power-law-tailed distributions.

## Figures and Tables

**Figure 1 entropy-23-00241-f001:**
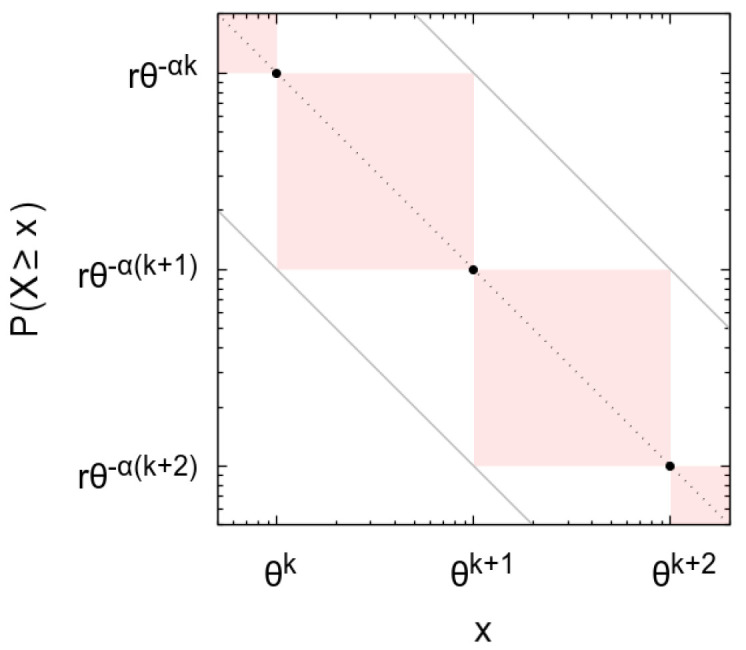
Representation in logarithmic scales of a complementary cumulative distribution having the law $P\left(X\geq x=\theta^{j}\right)=rx^{-\alpha }$, from a given *j* (black symbols). Values for $\theta^{j}\leq x\leq \theta^{j+1}$ can only occur in the red shaded areas, which define the bounds $r\theta^{-\alpha }x^{-\alpha }\leq P\left(X\geq x\right)\leq r\theta^{\alpha }x^{-\alpha }$ (gray lines).

**Figure 2 entropy-23-00241-f002:**
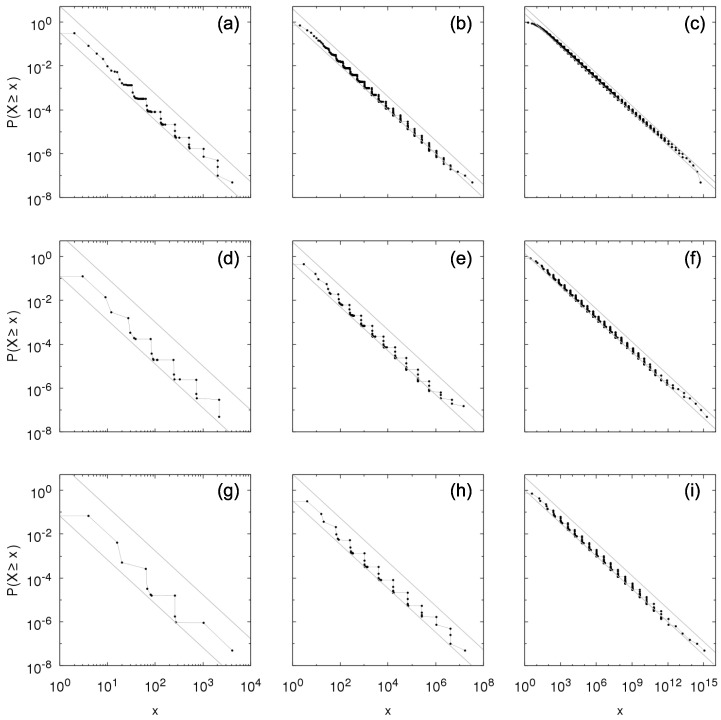
Numerical construction of the complementary cumulative distribution $P_{\left[ind\right]}^{\left(ind\right)}\left(X\geq x\right)$ of the Sigma-Pi X(t), *t* large, from the growth model with Bernoulli random variables for within- and between-independence with (**a**) θ=2, p=0.25; (**b**) θ=2, p=0.5; (**c**) θ=2, p=0.707; (**d**) θ=3, p=0.111; (**e**) θ=3, p=0.333; (**f**) θ=3, p=0.577; (**g**) θ=4, p=0.0625; (**h**) θ=4, p=0.25; and (**i**) θ=4, p=0.5. Gray lines show bounds for the distribution given by Equation (Equation 16): $\frac{p}{1-p}x^{-\alpha }\leq P_{\left[ind\right]}^{\left(ind\right)}\left(X\geq x\right)\leq \frac{1}{p(1-p)}x^{-\alpha }$, with α=2 in the left column, α=1 in the middle column and α=0.5 in the right column.

**Figure 3 entropy-23-00241-f003:**
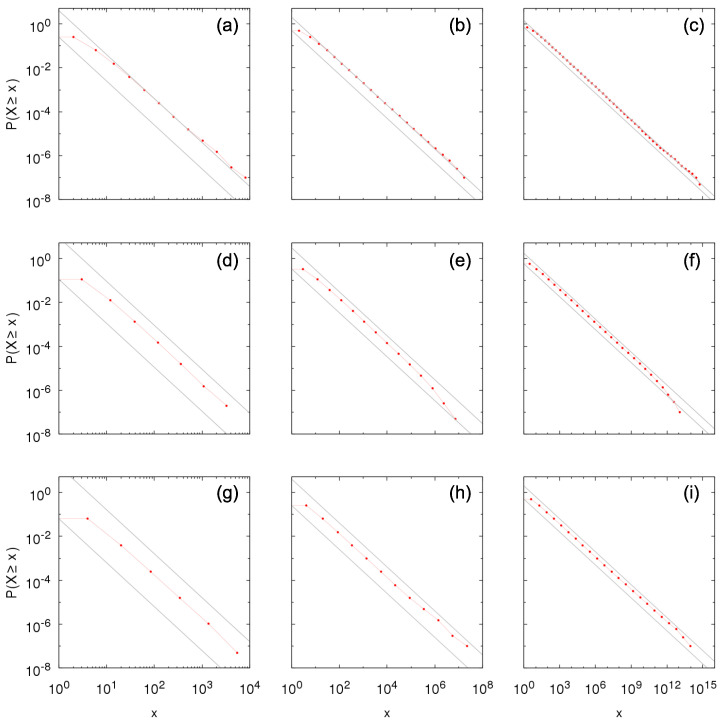
Numerical construction of the complementary cumulative distribution $P_{\left[ind\right]}^{\left(kes\right)}\left(X\geq x\right)$ of the Sigma-Pi X(t), *t* large, from the growth model with Bernoulli random variables for within-independence and Kesten between-dependence with (**a**) θ=2, p=0.25; (**b**) θ=2, p=0.5; (**c**) θ=2, p=0.707; (**d**) θ=3, p=0.111; (**e**) θ=3, p=0.333; (**f**) θ=3, p=0.577; (**g**) θ=4, p=0.0625; (**h**) θ=4, p=0.25; and (**i**) θ=4, p=0.5. Gray lines show bounds for the distribution given by Equation (Equation 18): $px^{-\alpha }\leq P_{\left[ind\right]}^{\left(kes\right)}\left(X\geq x\right)\leq \frac{1}{p}x^{-\alpha }$, with α=2 in the left column, α=1 in the middle column and α=0.5 in the right column.

**Figure 4 entropy-23-00241-f004:**
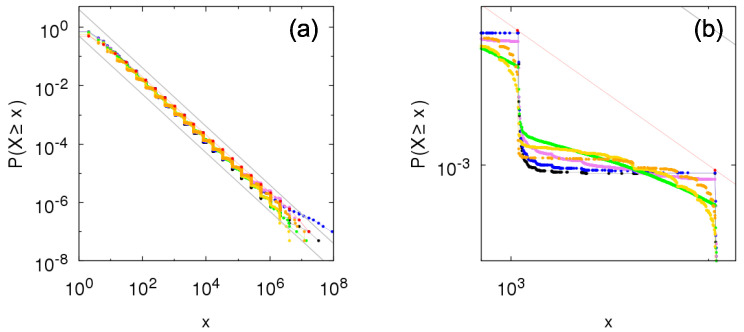
Numerical construction of the complementary cumulative distribution $P_{\left[ind\right]}^{\left(q\right)}\left(X\geq x\right)$ of the Sigma-Pi X(t), *t* large, from the growth model with Bernoulli random variables for within-independence and *q*-between-dependence with θ=2, p=0.5 and q=0 (red: Kesten between-dependence), q=0.1 (orange), q=0.25 (yellow), q=0.5 (green), q=0.75 (violet), q=0.9 (blue), and q=1 (black: between-independence). Gray lines show bounds for the distribution given by Equation (Equation 20): $px^{-\alpha }\leq P_{\left[ind\right]}^{\left(q\right)}\left(X\geq x\right)\leq \frac{1}{p(1-p)}x^{-\alpha }$, with α=1. (**b**) is a zoomed portion of (**a**).

**Figure 5 entropy-23-00241-f005:**
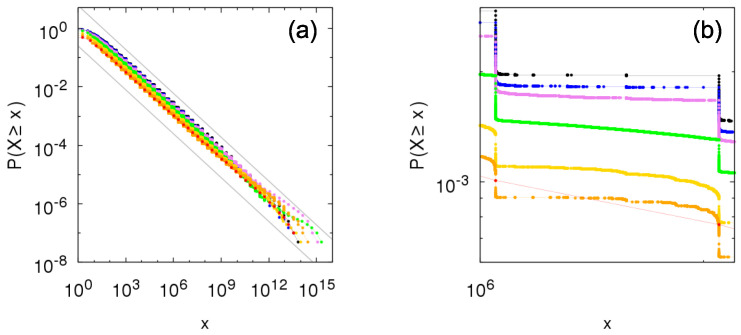
Numerical construction of the complementary cumulative distribution $P_{\left[\kappa =1\right]}^{\left(q\right)}\left(X\geq x\right)$ of the Sigma-Pi X(t), *t* large, from the growth model with Bernoulli random variables for κ-within-dependence, κ=1, and *q*-between-dependence with θ=2, p=0.5 and q=0 (red: Kesten between-dependence), q=0.1 (orange), q=0.25 (yellow), q=0.5 (green), q=0.75 (violet), q=0.9 (blue), and q=1 (black: between-independence). Gray lines show bounds for the distribution given by Equation (Equation 29): $p^{2}x^{-\alpha /2}\leq P_{\left[\kappa =1\right]}^{\left(q\right)}\left(X\geq x\right)\leq \frac{1}{p}\frac{1+p}{1-p}x^{-\alpha /2}$, with α=1. Panel (**b**) is a zoomed portion of panel (**a**).

**Figure 6 entropy-23-00241-f006:**
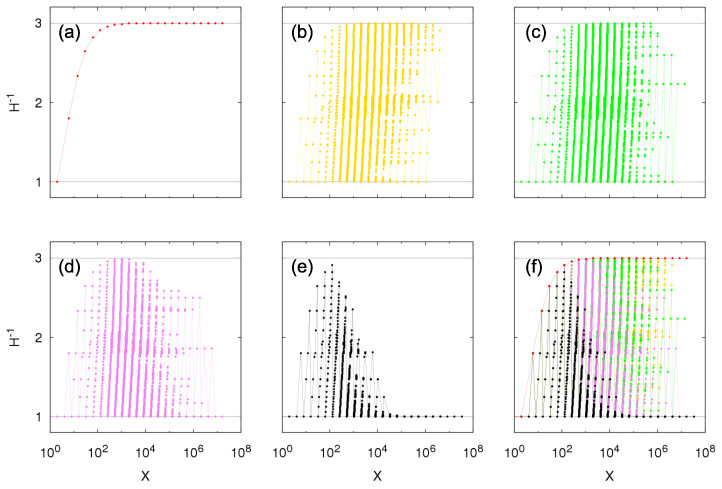
Numerical construction of the relationship between the inverse Herfindahl index $H^{-1}$ defined from Equation (Equation 30) and values of the Sigma-Pi X(t), *t* large, from the growth model with Bernoulli random variables for within-independence and *q*-between-dependence with θ=2, p=0.5 and (**a**) q=0 (Kesten between-dependence), (**b**) q=0.25, (**c**) q=0.5, (**d**) q=0.75, (**e**) q=1 (between-independence), and (**f**) all previous *q*. Horizontal gray lines correspond to $H^{-1}=1$.

**Figure 7 entropy-23-00241-f007:**
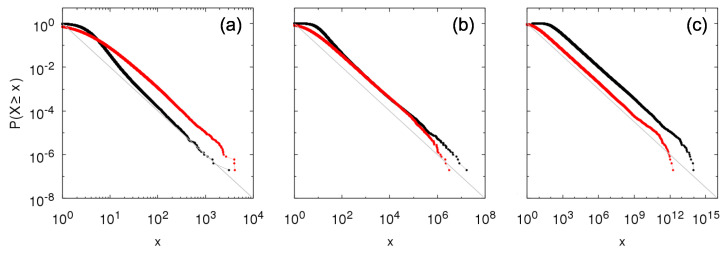
Numerical construction of the complementary cumulative distributions $P_{\left[ind\right]}^{\left(ind\right)}\left(X\geq x\right)$ and $P_{\left[ind\right]}^{\left(kes\right)}\left(X\geq x\right)$ of the Sigma-Pi X(t), *t* large, from the growth model with half-normal random variables for within-independence and between-independence (black) and Kesten between-dependence (red) with (**a**) θ=1, (**b**) θ=1.253, and (**c**) θ=1.479. Gray lines show $f\left(x\right)=x^{-\alpha }$, with α=2 in the left graph, α=1 in the middle graph and α=0.5 in the left graph.

**Figure 8 entropy-23-00241-f008:**
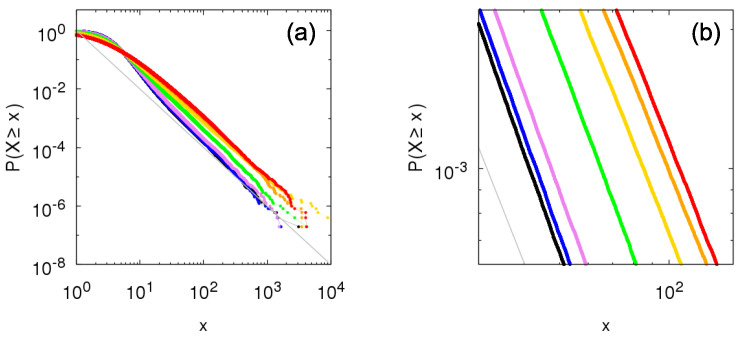
Numerical construction of the complementary cumulative distribution $P_{\left[ind\right]}^{\left(q\right)}\left(X\geq x\right)$ of the Sigma-Pi X(t), *t* large, from the growth model with half-normal random variables for within-independence and *q*-between-dependence with θ=1 and q=0 (red: Kesten between-dependence), q=0.1 (orange), q=0.25 (yellow), q=0.5 (green), q=0.75 (violet), q=0.9 (blue), and q=1 (black: between-independence). Gray line shows $f\left(x\right)=x^{-\alpha }$, with α=2. Panel (**b**) is a zoomed portion of panel (**a**).

**Figure 9 entropy-23-00241-f009:**
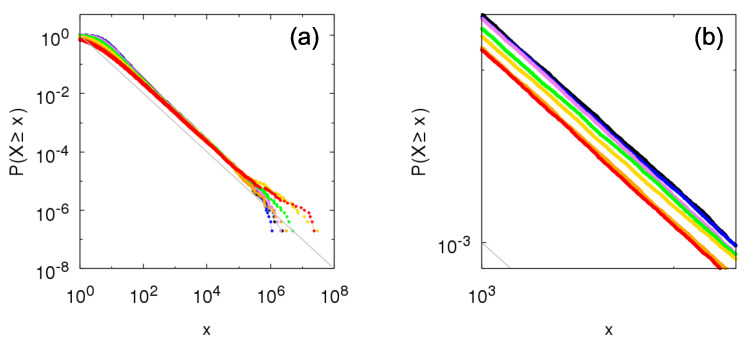
Numerical construction of the complementary cumulative distribution $P_{\left[\kappa =1\right]}^{\left(q\right)}\left(X\geq x\right)$ of the Sigma-Pi X(t), *t* large, from the growth model with half-normal random variables for κ-within-dependence, κ=1, and *q*-between-dependence with θ=1 and q=0 (red: Kesten between-dependence), q=0.1 (orange), q=0.25 (yellow), q=0.5 (green), q=0.75 (violet), q=0.9 (blue), and q=1 (black: between-independence). Gray line shows $f\left(x\right)=x^{-\frac{\alpha }{2}}$, with α=2. Panel (**b**) is a zoomed portion of panel (**a**).

**Figure 10 entropy-23-00241-f010:**
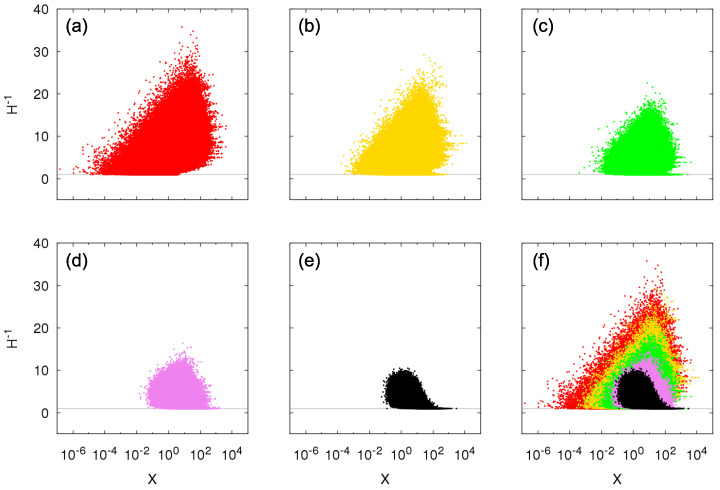
Numerical construction of the relationship between the inverse Herfindahl index $H^{-1}$ and values of the Sigma-Pi X(t), *t* large, from the growth model with half-normal random variables for within-independence and *q*-between-dependence with θ=1 and (**a**) q=0 (Kesten between-dependence), (**b**) q=0.25, (**c**) q=0.5, (**d**) q=0.75, (**e**) q=1 (between-independence), and (**f**) all previous *q*. Horizontal gray lines correspond to $H^{-1}=1$.

## Data Availability

Not applicable.

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
