# Peer review of "Sigma-Pi Structure with Bernoulli Random Variables: Power-Law Bounds for Probability Distributions and Growth Models with Interdependent Entities"

_entropy, 2021, doi:10.3390/e23020241_

Round 1
Reviewer 1 Report
In this article, the authors investigated the Sigma-Pi structure which consists of the sum of products of an increasing number of identically distributed random variables.
Most particularly, the Sigma-Pi structure with Bernoulli random variables are studied and noticed that its probability distribution is always bounded from below by a power-law function regardless of whether the random variables are mutually independent or duplicated.
This article is indeed a valuable addition to the existing literature on Power-Laws and its applicability in different field of research. My overall assessment, after having read the whole manuscript, is that the candidate article is acceptable for publication in the Entropy.
No comments for the authors.
Author Response
The authors thank the reviewer for the positive assessment and for the publication recommendation.
Reviewer 2 Report
Report on paper entitled “Sigma-Pi Structure with Bernoulli Random Variables: Power-Law Bounds for Probability Distributions and Growth Models with Interdependent Entities”:
In this paper, authors studied the Sigma-Pi structure with Bernoulli random variables and funded that its probability distribution is always bounded from below by a power-law function regardless of whether the random variables are mutually independent or duplicated and then authors investigate the case in which the asymptotic probability distribution has always upper and lower power-law bounds with the same tail-index. They illustrated the Sigma-Pi structure in the context of a simple growth model with successively born entities growing according to a stochastic proportional growth law, taking both Bernoulli, confirming the theoretical results, and half-normal random variables, for which the numerical results can be rationalized using insights from the Bernoulli case. They analyzed the interdependence among entities represented by the product terms within the Sigma-Pi structure, the possible presence of memory in growth factors and the contribution of each product term to the whole Sigma-Pi structure.
I do think that the paper is well written, but it needs minor revision:
1-The justification of using the Sigma-Pi structure with Bernoulli random variables should be added obviously in the introduction section.
2-Authors should motivate the paper according to:
- Why we need the Sigma-Pi structure?
- Why they considered the Sigma-Pi structure with Bernoulli random variables?
- Comparing the growth model with Bernoulli random variables and growth model with half-normal random variables.
3-More details about Equations (5-11) should be added.
4-An application to new real data set should be added along with its related graphical plots.
5-More useful comments related all Figures should be added.
6-The used codes “scripts” should be added into the appendix.
Reviewer 3 Report
this work consists in an investigation of the sum of products of an increasing number of identically distributed random variables, already discussed in [1], - a publication in Entropy, but I have no idea why this is called a « sigma-pi » process
its probability distribution is always bounded from below by a power-law function ; one could mention some information on the power law exponent in the abstract
the references might be considered to be a little bit biased; there are other papers based on ARCH/GARCH which describe growths ; why are they excluded ? why some are irrelevant ?
I observe that there is a novel discussion about the possible presence of memory in growth factors and the contribution of each product term to the whole Sigma-Pi structure.
The notations are a little bit weird : « Half-Normal Random Variables » ; I wonder when one is quarter normal.
In the future I wonder if instead of introducing an arbritrary threshold at simulation time one could not rather complicate Eq.(1) in introducing a nonlinear term, say à la Verhulst » in order to avoid infinite (and silly) growth. This suggestion much differs from the Sigma-Sigma-Pi structure: case proposed by the authors
Personally I don’t like the verb tenses at some places, like when the authors write « … We used half-normally distributed random variables to exemplify it. In particular, we observed … ». This is a typical example, others are hidden in the text, about the wrong use of this past tense. The present tense is fully more correct.
Thus,
The present paper although a complement to a previous paper is of interest, since the content and study are noveL. The paper is of interest because of its content. The mathematical formulation is likely correct. The « applications » are sufficiently well hinted to, but I don't know when I can use this process The use of the grammar is different from mine, - but so what ?
I suggest to accept the paper, and let the authors adapt remarks by any reviewer at galley proof time.
